# Empirical evaluation of the association between daily living skills of adults with autism and parental caregiver burden

**Christina N. Marsack-Topolewski[1], Preethy Sarah Samuel[2]\*, Wassim Tarraf[2,3]**

**1** School of Social Work, Eastern Michigan University, Ypsilanti, Michigan, United States of America,
**2** Occupational Therapy Program, Department of Health Care Sciences, Wayne State University, Detroit, Michigan, United States of America, **3** Institute of Gerontology, Wayne State University, Detroit, Michigan, United States of America

\* ak0341@wayne.edu

## Abstract

### Background

Despite the joy of parenting, the burden of daily caregiving for children with autism spectrum disorders (ASD) can be overwhelming and constant. Parents can expect to provide enduring care for their children with ASD. Given that the majority of individuals with autism spectrum disorders (ASD) remain in their family homes well into adulthood, often the need for assistance with activities of daily living (ADLs) is placed on parents. Providing ongoing support to adult children who have difficulty with completing ADLs can increase parental caregiving demands. Therefore, the purpose of this study was to examine the relationship between the ability of adults with ASD to perform ADLs with parental perceptions of caregiver burden.

### Methods

Quantitative analysis of cross-sectional multi-state data gathered electronically using Qualtrics from 320 aging parents of adults with ASD was conducted. Regression models were fit to examine the association of ADL challenges with total caregiver burden and its four domains (emotional, financial, time dependence, and developmental).

### Results

Parental perceptions of caregiver burden decreased, particularly time dependence and developmental burden, when adult children were less dependent in ADLs, even after adjusting for parental health and behavioral challenges.

### Conclusions

Findings support the need for family-centered interventions to improve the capacity of adults with ASD to perform ADLs independently.

**Data Availability Statement:** All relevant data are within the manuscript and its Supporting Information files.

**Funding:** This research was funded in part by dissertation research funding from Wayne State University's Graduate School and the School of Social Work that enabled the data collection. The funders had no role in study design, data collection and analysis, decision to publish, or preparation of the manuscript.

**Competing interests:** The authors have declared that no competing interests exist.

## Introduction

Estimates of the prevalence of autism spectrum disorders (ASD) worldwide indicated that of the 6.6 million individuals with this diagnosis, 5.3 million are adults and 1.3 million are children and adolescents aged 3–19 years [1]. As individuals with ASD reach adulthood, many continue to need some form of assistance from their parents. A small proportion of adults with ASD live independently when compared to adults with other types of disabilities [2]. Past research indicated that about 50–80% of adults with ASD were living with their parents [2–4]. These parents typically do not experience the "empty nest" when their children leave home to pursue post-secondary education, employment, or marriage [5].

Co-residing with adult children can be beneficial for aging parents because of the reciprocal nature of caregiving [6]. The other benefits of caring for children with ASD include positive personality changes in family members such as becoming more patient, less judgmental, and gaining resiliency using problem-focused coping strategies [7,8]. Despite the joy of parenting [9], providing daily care for adult children with ASD can be overwhelming and constant, contributing to caregiver burden [10].

Many adults with ASD continue to need support with activities of daily living (ADLs) across their lifespan [11,12]. Given that the majority of individuals with ASD remain in their family homes well into adulthood, the need for assistance with ADLs often is placed on parental caregivers [2]. There is evidence to suggest that parental well-being is associated with the daily living skills of children with ASD [13,14]. Providing ongoing support for adult children with ASD who have difficulty performing ADLs independently can increase the level of caregiver burden experienced by their aging parents [12,15]. Much of the evidence demonstrating the association of caregiver burden with functional independence in caregiving literature is based on research conducted among adult-onset conditions, such as stroke and dementia [16,17] and unspecified intellectual/ developmental disabilities [18,19]. In comparison, there is sparse research on the dynamic association between caregiver burden and functional independence of children with ASD, particularly among adolescents and adults [20–22].

### Caregiver burden

Caregiver burden is a multifactorial construct. In this paper, caregiver burden experienced by aging parents of adults with ASD was conceptualized to comprise four domains- emotional, developmental, time dependence, and financial burden, based on adult caregiving literature [23,24]. Parents experience *emotional burden*, which includes negative emotions such as sadness, grief, depression, anxiety, guilt, and blame [25–27]. The *developmental* aspect of burden pertains to parents' perceptions of being "off-time" in their development when compared to their same-aged peers. Extant evidence indicated that parental caregivers were more likely to report missing out on a normal way of life, experience social disconnectedness from relatives and friends, and feel unsupported by systems and society at large [28,29]. *Time dependence* is the perception of burden secondary to the time restrictions faced by parents because of the enduring and lifelong demands of caregiving for adult children [24,30]. Time devoted to caregiving can reduce opportunities to associate with friends, family members, and spouses, and can restrict parent caregivers' potential for pursuing career, leisure, and community interaction opportunities [7,31]. *Financial burden* encompasses ongoing economic costs that families contend with to ensure that all family members, including the adult child with ASD, have access to age-appropriate health care, educational, and career opportunities. Parents often have to reduce working hours, forgo promotions, and higher educational pursuits because of caregiving demands that can affect family income and retirement benefits [31,32].

## Predictors of caregiver burden

Caregiver burden is influenced by individual- and family-level factors. Individual-level factors include the clinical and socio-demographic characteristics of the individual with ASD and their caregivers. The core clinical traits of ASD (*e.g.*, ability to communicate and socialize) influences an individual's functional independence, placing differential demands on parental caregivers [13,30,33]. Many socio-demographic characteristics of parents also can influence their perceptions of the burden associated with caregiving. Aging parents, who are themselves experiencing age-related health challenges while supporting their adult children with ASD, are likely to experience greater caregiver burden, than younger parents [6,34]. In addition, many aging parents also may be providing care to other family members such as another child, spouse, or parent [35]. Recent estimates indicated that close to a quarter of informal family caregivers in the United States are providing care for more than one family member [36]. Compound caregiving for multiple individuals in aging families can also increase a parent's perceived caregiver burden [37,38].

**Activities of daily living.**   According to the *Occupational Therapy Practice Framework* [39], ADLs are divided into two types, basic and instrumental ADLs. Basic ADLs are oriented towards taking care of one's own body (*e.g.*, eating, grooming, bathing, toileting) and are fundamental to participation in daily life. Instrumental ADLs that support participation in complex daily tasks (*e.g.*, meal preparation, house cleaning, shopping) are oriented to interacting with the environment. Independence in ADLs is associated with positive outcomes in adulthood, such as post-secondary education, employment, and independent living [15,40–42]. Individuals with ASD often face internal and external challenges in mastering ADLs.

The internal challenges refer to individual characteristics of the person with ASD such as the physical, cognitive, emotional and psychosocial impairments. These challenges contribute to a low internal drive and motivation to be independent, particularly in instrumental ADLs such as money management and transportation that are often optional and can be delegated to others [43–45]. Impaired social communication skills and behavioral challenges, ranging from stereotypical, repetitive, self-injurious to very aggressive behaviors, often require supervision and redirection from others and also foster dependence on family members to perform their ADLs [46–48]. Although it is well-established that children with ASD benefit from interventions targeted to improve their daily living skills [49–52], there is a paucity of interventions focused on individuals with ASD across the lifespan and their family members [53,54].

The external challenges in fostering functional independence of individuals with ASD include environmental issues and the systemic challenges in the access and use of supportive services. Service utilization research has demonstrated that occupational therapy, which focuses on enhancing the functional independence of all individuals across the lifespan, was one of the least utilized therapeutic services for individuals with ASD [55,56]. Turcotte et al. [55] reported that parents of adults with ASD were least likely to use occupational therapy when compared to parents of younger children in preschool, elementary and middle/high school. Infrequent use of occupational therapy is concerning because adults with ASD are at risk of losing functional independence.

Results from a 10-year longitudinal study of 397 adolescents and adults with ASD indicated that independence in ADLs typically improved during adolescence and emerging adulthood, plateaued in the late 20s, and declined in the early 30s [12]. Importantly, adults with ASD were unable to complete more than 30% of their ADLs independently. These findings highlight the critical need for ongoing support in adulthood to maintain independence in basic ADLs and gain independence in instrumental ADLs. Parental caregivers who provide daily supports are

also in need of supportive interventions with a primary focus on fostering mastery of ADLs of adults with ASD as demonstrated in other disability populations [57].

## Study purpose

The purpose of this study was to examine the relationship between the ability of adults with ASD to perform ADLs and parental perceptions of caregiver burden. It was hypothesized that higher dependence in ADLs would be associated with a greater perception of caregiver burden after adjusting for factors in the caregiving context.

## Materials and methods

### Research design

Using a web-based survey, cross-sectional data was collected electronically from a convenience sample of parents of adult children with ASD from multiple states in the United States.

**Procedures.** Approval was obtained from Wayne State University's Institutional Review Board (IRB# 012615B3X) prior to beginning data collection. Multiple strategies were used to recruit participants including (1) contacting organizations that interfaced with families of adults with ASD, (2) communicating with professionals (*e.g.*, social workers, educators, transition coordinators), (3) making face-to-face contact with potential participants at support groups and other family-focused events, and (4) using snowball sampling with already recruited participants. The organizations that were included in the data collection included: (a) Autism Alliance of Michigan, (b) Autism Society of Macomb/St. Clair, Michigan, (c) Autism Society of Oakland County, Michigan, (d) Autism Society of Wisconsin, (e) Judson Center, Michigan, (f) Milestones Autism Resources, Ohio, (g) Shelby County Regional Special Education Parent Teacher Association, Tennessee, (h) the Autism Program of Illinois. These organizations and professionals agreed to place the survey link in their newsletters, websites, flyers, and to publicize through word of mouth. Potential participants received information regarding the study, including the link to the electronic web-based survey [58] that was developed using the Qualtrics software platform.

**Power analysi.** Power analysis using G*Power 3.1 [59] was undertaken before starting data collection to determine the appropriate sample size. The results indicated the need for a sample of 242 participants to conduct multiple regression analyses with six independent variables, using blocks, at an alpha level of 0.05, and a power of 0.80.

### Participants

Participants had to be at least 50 years of age, able to read and comprehend English, have access to a computer for the web-based survey, and self-report being a parent of at least one adult (aged 18 or above) with ASD to be included in the study. Eligible participants were asked to read the information sheet and if they were willing to participate in the study, they were directed to complete the online survey. A total of 353 self-administered surveys were submitted to the website. This number was reduced by 33 participants, including 23 who did not provide a sufficient number of responses and 10 who did not meet the inclusion criteria because the child with ASD had not reached the age of 18 years or the parent was younger than 50 years of age. The final sample of older parents included 320 respondents who met all criteria for inclusion.

The majority of the participants (*n* = 212, 66.3%) were aged 50–59 years, while the remaining (*n* = 108, 33.7%) were aged 60-years or older. Most of the participants were female (*n* = 259, 82.0%), Caucasian (*n* = 289, 90.9%), reported being married (*n* = 254, 79.2%), and had higher levels of education (*n* = 223, 69.7% with a bachelor's degree or higher). The

majority (*n* = 204, 63.8%) were working either full-time, part-time, or were self-employed, and more than half (*n* = 185, 59.1%) reported an annual family income greater than $60,000 (Table 1).

## Variables of interest

**Dependent variables.**   Perceptions of parental caregiver burden were assessed using 15 items from the Caregiver Burden Inventory (CBI) and three items from the Caregiver Reaction Assessment (CRA). The CBI was developed to measure the perceptions of caregiving for individuals with Alzheimer's disease [24]. The CRA developed by Given et al. [23] has been used internationally to examine caregivers' reactions to providing care for family and friends with a variety of chronic illnesses. Items from these tools that were deemed relevant to assessing perceptions of parent caregivers of adults with ASD were incorporated into this study. Permission from the authors of the surveys was obtained before modifying the language to be relevant to use with parents of adult children.

**Time dependence burden.**   The five CBI items used to measure the perceived restrictions on the caregiver's time were (1) He/she needs my help to perform many daily tasks, (2) He/she is dependent on me, (3) I have to watch him/her with many basic functions, (4) I have to help him/her with many basic functions, (5) I don't have a minute's break from his/her chores. Parents rated these items using a 5-point rating scale where higher scores indicated greater time dependence burden (1 = never, 5 = always). The internal consistency ($\alpha$ = 0.91) of this subscale for this sample was excellent.

**Developmental burden.**   The five CBI items used to measure parents' feelings of being "off-time" in their development with respect to their peers were (1) I feel that I am missing out on life, (2) I wish I could escape from this situation, (3) My social life has suffered, (4) I feel emotionally drained due to caring for him/her, (5) I expected that things would be different at this point in my life. Parents rated these items using a 5-point rating scale where higher scores indicated greater developmental burden (1 = never, 5 = always). The internal consistency ($\alpha$ = 0.91) of this subscale for this sample was excellent.

**Emotional burden.**   The five CBI items used to measure the negative feelings associated with providing care for adult children with ASD were: (1) I feel embarrassed over his/her behavior, (2) I feel ashamed of him/her, (3) I resent him/her, (4) I feel uncomfortable when I have friends over, (5) I feel angry about my interactions with him/her. Parents rated these items using a 5-point rating scale where higher scores indicated greater emotional burden (1 = never, 5 = always). The internal consistency ($\alpha$ = 0. 88) of this subscale for this sample was good. Overall, the internal consistency of these three 5-item adapted CBI subscales used in this study was similar to reports from past studies [60].

**Financial burden.**   The three CRA items used to measure impact of family finances on caregiving were (1) Financial resources are adequate. (2) It is difficult to pay for my adult child with ASD. (3) Caring for my adult child with ASD puts a financial strain on me. Parents rated these items using a 5-point Likert scale (1 = strongly disagree to 5 = strongly agree) with higher scores indicating greater financial burden. The internal consistency ($\alpha$ = 0.87) of this scale using this sample was good and higher than values ($\alpha$ = .62) reported in previous studies of caregivers caring for older adults [23].

**Parental caregiver burden.**   The mean computed from the four subscales represented the total burden experienced by parents of adults with ASD, where higher scores indicated greater burden. The internal consistency of the 18-item burden scale was excellent ($\alpha$ = .91).

**Independent variables.**   The parents rated their adult child's ability to perform 11 ADLs: personal hygiene/ grooming, eating, dressing, bathing, toileting, mobility, community

**Table 1.  Descriptive statistics.**

| Parental Caregiver Characteristics | | N | % | Child Characteristics | | N | % |
|---|---|---|---|---|---|---|---|
| **Age** | | | | **Gender** | | | |
| | 50–59 years | 212 | 66.3 | | Male | 257 | 80.3 |
| | 60 years or above | 108 | 33.8 | | Female | 63 | 19.7 |
| **Gender** | | | | **Living arrangement** | | | |
| | Males | 57 | 17.8 | | Live with parents | 248 | 77.5 |
| | Females | 259 | 80.9 | | Live independently with support | 35 | 10.9 |
| | Missing | 4 | 1.3 | | Live independently | 16 | 5.0 |
| **Race** | | | | | Live in a group home | 11 | 3.4 |
| | White | 289 | 90.3 | | College | 6 | 1.9 |
| | Other | 29 | 9.1 | | Other | 4 | 1.3 |
| | Missing | 2 | 0.6 | **Employment status** | | | |
| **Marital status** | | | | | Employed full-time | 21 | 6.6 |
| | Married/Cohabitating | 64 | 20.0 | | Employed part-time | 59 | 18.4 |
| | Not Married | 254 | 79.4 | | Working with a job coach | 41 | 12.8 |
| | Missing | 2 | 0.6 | | Employed in a sheltered workshop | 13 | 4.1 |
| **Employment status** | | | | | College | 16 | 5.0 |
| | Employed full-time | 118 | 36.9 | | High school/ post-secondary | 52 | 16.3 |
| | Employed part-time | 79 | 24.7 | | Unemployed | 53 | 16.6 |
| | Self-employed | 7 | 2.2 | | Unable to work | 58 | 18.1 |
| | Not employed | 116 | 36.3 | | Other | 6 | 1.9 |
| **Educational attainment** | | | | | Missing | 1 | 0.3 |
| | High school or less | 11 | 3.4 | **Educational attainment** | | | |
| | Some college | 86 | 26.9 | | Attending university or college | 29 | 9.1 |
| | Bachelor's degree | 100 | 31.3 | | Attending community college | 24 | 7.5 |
| | Master's degree or higher | 123 | 38.4 | | Attending vocational training program | 13 | 4.1 |
| **Household income** | | | | | Attending public school | 75 | 23.4 |
| | $40,000 or below | 38 | 11.9 | | Unable to attend educational program | 27 | 8.4 |
| | $40,001 to $60,000 | 42 | 13.1 | | Other | 148 | 46.3 |
| | $60,001 to $100,000 | 82 | 25.6 | | Missing | 4 | 1.3 |
| | More than $100,000 | 103 | 32.2 | **Communication** | | | |
| | Missing | 55 | 17.2 | | Uses device | 4 | 1.3 |
| **Compound caregiving** | | | | | Very little meaningful communication | 17 | 5.3 |
| | No | 206 | 64.4 | | Basic needs & wants | 33 | 10.3 |
| | Yes | 110 | 34.4 | | Needs & wants in some meaningful way | 29 | 9.1 |
| | Missing | 4 | 1.3 | | Limited range of topics meaningfully | 112 | 35.0 |
| **Caregiver health** | | | | | A wide variety of topics meaningfully | 124 | 38.8 |
| | Poor | 6 | 1.9 | | Missing | 1 | 0.3 |
| | Fair | 39 | 12.2 | **Behavioral** | | | |
| | Good | 101 | 31.6 | | Requires 24-hour supervision | 16 | 5.0 |
| | Very good | 108 | 33.8 | | Engages in self-injurious behavior | 19 | 5.9 |
| | Excellent | 59 | 18.4 | | Requires substantial prompting, direction | 108 | 33.8 |
| | Missing | 7 | 2.2 | | Requires minimal prompting, direction | 160 | 50.0 |
| | | | | | Typical, age appropriate adult behavior | 14 | 4.4 |
| | | | | | Missing | 3 | 0.9 |

*(Continued)*

**Table 1.** (Continued)

| Parental Caregiver Characteristics | | | | Child Characteristics | | | |
|---|---|---|---|---|---|---|---|
| | | N | % | | | N | % |
| | | | | Social (%) | | | |
| | | | | | Socializes with family only | 82 | 25.6 |
| | | | | | Is not able to make friends | 32 | 10.0 |
| | | | | | Would like friendships, but has no friends | 73 | 22.8 |
| | | | | | Is part of a social group | 88 | 27.5 |
| | | | | | Has friends, goes out with others in the community | 43 | 13.4 |
| | | | | | Missing | 2 | 0.6 |

participation, bed making or cleaning, cooking simple meals, managing money, and public transportation, using a 4-point rating scale (4 = requires total assistance to perform the task to 1 = person with ASD can perform the task independently). The mean scores were computed, such that higher total scores indicated greater dependence in ADLs. The internal consistency of this 11-item scale was excellent ($\alpha$ = .90).

**Parent characteristics.** Two caregiving characteristics associated with the parents that were included in this study were self-reported health and compound caregiving (*i.e.*, caring for other members in the family). Self-reported parents' health was measured using a single item with 5-point scale ranging from poor to excellent, with higher scores indicating better health. Compound caregiving was measured as a dichotomous variable, with participants indicating if they were providing care for another person(s) in addition to their adult child with ASD. A total of 110 parents reported providing care for another person. About half of these parents were caring for another child (50.0%), parents (23.1%), spouse (20.0%), and others (6.9%).

**Adult child with ASD characteristics.** Three aspects of the core traits of ASD examined in this study were: communication, social skills, and behavioral challenges of the adult children. Parents rated the communication ability of their adult child using a 6-point rating scale (1 = communicates with a device, 2 = very little meaningful communication, 3 = communicates basic needs and wants, 4 = communicates needs and wants in some meaningful way, 5 = communicates in a limited range of topics in a meaningful way, and 6 = communicates a wide variety of topics in a meaningful way). This indicator was included as a nominal variable in the model using category 4 "communicates needs and wants in some meaningful way" as the reference category. Parents rated their adult child's ability to engage in social interaction using a 5-point rating scale (1 = socializes with family only, 2 = is not able to make friends, 3 = would like friendships, but has no friends, 4 = is part of a social group, and 5 = has a group of friends, goes out with others in the community), where higher scores indicated greater social interaction. Parents rated the behavioral challenges of their adult child using a 5-point rating scale (1 = requires 24-hour supervision to manage behavior, 2 = engages in self-injurious behavior and/or dangerous/ aggressive behavior, 3 = requires substantial prompting, direction, or redirection, 4 = requires minimal prompting, direction, or redirection, and 5 = typical, age appropriate adult behavior), where higher scores indicated greater use of age-appropriate behaviors. Each of these scales were developed for this study based on literature review.

## Data analysis

First, frequency distributions and descriptive statistics were used to summarize all variables of interest (Table 1). Second, prior to running regression models, pairwise correlation analyses

were used to examine the bivariate relationships between the independent and dependent variables. Correlation plots were generated (S1 Fig) using the R software package [61,62]. To characterize the form of the association between the primary independent variable (dependence in ADLs of the adult child with ASD) and each of the dependent variables (caregiver burden and the four domains) linear, quadratic, and non-parametric Lowess functions [63] were fitted and plotted (S2 Fig). Based on these estimates it was determined that the linear form provides optimal fit (also see sensitivity analyses below). As such, linear associations were adopted and presented as primary findings in this study.

For each dependent variable three linear regression models were fit to examine (1) crude, (2) caregiver characteristics, and (3) adult with ASD characteristics adjusted models. The aim of these incremental adjustments was to determine the extent to which the associations between caregiver burden and dependence in ADLs would be attenuated by these adjustments. Table 2 includes the unstandardized beta coefficients derived from these models. To maintain parsimony, variables with bivariate associations significant at $p < 0.05$ were retained for use as covariates in the regression models (however, see sensitivity analyses below). S3 Fig summarizes the marginal means plots for each of the caregiver burden outcomes over the dependence in ADLs continuum, and their 95% confidence intervals. These plots help visualize the reported associations and the reductions in magnitudes resulting from covariates adjustments [64].

Several sensitivity analyses were conducted to ensure robustness of the primary findings. Sensitivity results are included as supplementary material to the manuscript. First, several

**Table 2. Linear associations between dependence in ADLs and caregiver burden[1].**

| | Total burden | | | Emotional burden | | | Developmental burden | | | Time dependence burden | | | Financial burden | | |
|---|---|---|---|---|---|---|---|---|---|---|---|---|---|---|---|
| | M1 | M2 | M3 | M1 | M2 | M3 | M1 | M2 | M3 | M1 | M2 | M3 | M1 | M2 | M3 |
| **ADL dependence** | 0.47*** | 0.43*** | 0.30*** | 0.06 | 0.04 | -0.06 | 0.48*** | 0.42*** | 0.27* | 1.20*** | 1.18*** | 0.98*** | 0.12* | 0.09 | 0.02 |
| **Compound caregiving** | | 0.07 | 0.09 | | 0.01 | 0.05 | | 0.15 | 0.18 | | -0.02 | 0.02 | | 0.12 | 0.12 |
| **Caregiver health** | | -0.13*** | -0.14*** | | -0.08 | -0.10* | | -0.25*** | -0.27*** | | -0.10** | -0.09** | | -0.09** | -0.11** |
| **Communication[2]** | | | | | | | | | | | | | | | |
| Uses device | | | 0.20 | | | 0.59 | | | 0.01 | | | 0.18 | | | 0.01 |
| Very little communication | | | 0.14 | | | 0.42 | | | -0.07 | | | 0.08 | | | 0.14 |
| Basic needs & wants | | | 0.04 | | | 0.16 | | | 0.14 | | | -0.04 | | | -0.11 |
| Needs/wants in some meaningful way | | | ref | | | ref | | | ref | | | ref | | | ref |
| Limited range of topics | | | 0.32** | | | 0.60*** | | | 0.43* | | | 0.09 | | | 0.15 |
| Wide variety of topics | | | 0.25* | | | 0.43* | | | 0.42 | | | -0.01 | | | 0.18 |
| **Behavior** | | | -0.24*** | | | -0.20** | | | -0.33*** | | | -0.26*** | | | -0.16** |
| **Social** | | | -0.02 | | | -0.05 | | | -0.04 | | | 0.00 | | | 0.00 |
| **Intercept** | 1.81*** | 2.30*** | 3.24*** | 1.74*** | 2.04*** | 2.70*** | 2.03*** | 2.96*** | 4.24*** | 0.66*** | 1.04*** | 2.21*** | 2.82*** | 3.15*** | 3.79*** |
| **R²** | 0.25 | 0.30 | 0.38 | 0.00 | 0.01 | 0.10 | 0.10 | 0.16 | 0.21 | 0.65 | 0.69 | 0.73 | 0.02 | 0.05 | 0.09 |
| **N** | 320 | 309 | 303 | 320 | 309 | 303 | 320 | 309 | 303 | 320 | 309 | 303 | 315 | 307 | 301 |

**Notes**: M1: Crude; M2: Adjusted for caregiver characteristics; M3: Fully adjusted.

*$p < 0.05$

**$p < 0.01$

***$p < 0.001$.

[1]Results are based on incrementally adjusted linear regression models. Unstandardized beta coefficients are presented in the Table.

[2]Communication was modeled as a nominal variable using "Communicates needs and wants in some meaningful way" as the reference category.

power transformations of the primary exposure (dependence in ADLs), including higher poly-nomial (quadratic, and cubic) and natural-log functions were considered and tested (S1 Table). AIC and BIC fit statistics showed no consistent evidence for improvement in fit (rela-tive to linear) as a result of these transformations. However, the covariates adjusted models for the quadratic and cubic transformations were used to calculate and plot the marginal means derived from these models, and their 95% confidence intervals, to allow interested readers to compare the estimates across linear, quadratic and cubic fit (S2 Table and S4 Fig). Second, to test for any possible attenuations in the primary results derived from the linear regression described above (regression model 3) a fourth model was fitted to adjust for the full set of covariates described in Table 1. The estimated marginal means derived from these models plotted in S5 Fig allow readers to visualize and compare these estimates relative to the primary reported findings. The data used for these analyses and the associated codebook are available for review in S1 and S2 Data respectively.

## Results

### Description of adult children with ASD

The majority (80.3%) of adults with ASD represented in this study were male, with a mean age of 25.1 years ($SD = 7.09$). Most (77.5%) were living with their parents, and about 42% were working for pay (full-time, part-time, with a job coach, or in a sheltered workshop). About 23.4% of the adult children were attending public schools, 16.6% were attending a university or community college, 4.1% were attending a vocational training program. Most adults with ASD (73.8%) could communicate on topics in a meaningful way. About half (50.3%) required minimal prompting, direction, or redirection for their behavior, and 4.4% had typical age-appropriate adult behavior, while 11% of parents reported serious challenging behaviors. About 42% had friends in the community or were part of a social group, while the remaining had no friends outside of their family (Table 1).

The mean scores for the dependence in ADLs in descending order (higher scores indicate greater dependence on parents) were: money management ($M = 2.92$), $SD = 1.13$), taking pub-lic transportation ($M = 2.80$, $SD = 1.26$), cooking simple meals ($M = 2.31$, $SD = 1.14$), commu-nity engagement ($M = 2.11$, $SD = 1.07$), simple cleaning e.g. bed making ($M = 2.10$, $SD = 1.0$), grooming ($M = 1.77$, $SD = 0.93$), bathing ($M = 1.51$, $SD = 0.91$), dressing ($M = 1.33$, $SD = 0.67$), eating ($M = 1.25$, $SD = 0.57$), functional mobility ($M = 1.26$, $SD = 0.65$), toileting ($M = 1.28$, SD = 0.65), and with an overall mean of 1.86 ($SD = 0.68$).

### Parental caregiver burden description

The mean for total burden as reported by the participants was 2.67 ($SD = 0.63$), when mea-sured on a 5-point rating scale, with higher scores indicating greater caregiver burden. The domain level scores indicated that the highest type of burden was financial ($M = 3.05$, $SD = 0.65$), followed by developmental ($M = 2.92$, $SD = 1.05$), followed by time dependence ($M = 2.88$, $SD = 1.01$), and emotional ($M = 1.84$, $SD = 0.76$).

### Correlation analysis

The results of pairwise correlation analysis summarized in S1 Fig indicate the direction and magnitude of the associations between the variables of interest. Total caregiver burden was positively associated with dependence in ADLs ($r = 0.50$, $p < 0.001$) and compound caregiving ($r = 0.13$, $p = 0.02$). The inverse associations of burden with adult child's ASD characteristics indicated that parents of adult children with fewer behavioral ($r = -0.48$, $p < 0.001$), and social

challenges ($r$ = -0.18, $p$ = .002) reported less caregiver burden. The inverse association of burden with health ($r$ = -0.27, $p$ <0.001) imply that parents with lower quality of health perceived higher levels of caregiver burden. The magnitude of these associations was small for emotional and financial burden.

## Regression analyses

First, unadjusted bivariate linear regression models showed that the associations between dependence in ADLs and caregiver burden (with the exception of emotional burden) were all statistically significant and in the expected direction, indicating that parents of adult children who needed to provide more assistance for ADLs were more likely to perceive greater caregiver burden. The estimated unstandardized coefficients were $b$ = 0.47 ($p$ < 0.001), $b$ = 0.48 ($p$ < 0.001), $b$ = 1.2 ($p$ < 0.001), and $b$ = 0.12 ($p$ = 0.023) for total, developmental, time dependence, and financial burden, respectively. The explained variances shown by the estimated $R^2$ were 25%, 10%, 65%, and 2%, respectively (Table 2).

Second, adjusting for parent characteristics attenuated the associations (unstandardized beta coefficients) between dependence in ADLs and total caregiver burden ($b$ = 0.43; [$p$ < 0.001]), developmental burden ($b$ = 0.42; [$p$ < 0.001]), and time dependence burden ($b$ = 1.18; [$p$ <0.001]) by 9%, 13%, and 2%, respectively. Adjustment for parent characteristics led to an additional 5%, 6%, and 4% absolute increase in level of explained variance for these types of burden, respectively. The association between dependence in ADLs and financial burden became statistically insignificant at this step.

Finally, adjusting for behavioral challenges further attenuated, but did not completely explain, the magnitudes of the associations (unstandardized beta coefficients) between dependence in ADLs and total burden ($b$ = 0.24; [$p$ <0.001]), developmental burden ($b$ = 0.27; [$p$ < 0.05]), and time dependence burden ($b$ = 0.98; [$p$ < 0.001]) by an additional 30%, 36%, and 17%, respectively. Adjustment for characteristics of the adult child with ASD led to an additional 8%, 5%, and 4% absolute increase in the levels of explained variance for these burdens, respectively. Plots of the marginal means derived from these models to help visualize the associations are presented in S3 Fig. Additionally, as expected, the sensitivity analyses (S5 Fig) showed no evidence for attenuations or change in the associations between the caregiver burden outcomes and ADLs from additional controls for the covariates detailed in Table 1.

## Discussion

Three main findings that emerged from investigating the cross-sectional data gathered from 320 aging parents of adult children with ASD. *First*, the linear association found between dependence in ADLs and total caregiver burden imply that parents reported greater caregiver burden when their adult children needed more assistance to complete their ADLs. This association was robust to adjustment for characteristics of the parent (i.e., parental health) and the adult child with ASD (i.e., behavioral challenges). This finding adds to a growing body of literature highlighting the importance of the functional independence of adolescents and adults with ASD on parental well-being [20,65,66]. Past literature focused on younger families of children with ASD indicated that behavioral problems have a greater influence on parental well-being than the functional abilities of children with ASD [13]. There is a noticeable lack of evidence on the consequences of dependence in ADLs of adult children with ASD and parental perceptions of caregiver burden. These findings fill this gap in the parental caregiving literature.

*Second*, it was found that the dependence in ADLs of adult children with ASD primarily influenced two types of caregiver burden. Parents of adults who were more independent in

completing their ADLs reported less developmental and time dependence burden. The adult child's dependence in ADLs was particularly associated with time dependence burden of their parental caregivers. The time demands faced by parents who have to assist their adult children with ADLs can affect their time commitments to career, leisure, or community engagements. While time dependence and developmental burden have been explored in other fields (*e.g.* dementia), there has been less thorough investigation in the field of intellectual and developmental disabilities [67]. Past research on families of children with disabilities largely focused on describing the adverse effects faced by mothers who are hard pressed for time because of the daily care of their children with ASD leading to poorer health, financial, and community participation outcomes when compared to mothers of children with other disabilities and neurotypical children [31]. Parents who are less involved with everyday caregiving, such as ADLs, were more likely to find the time to engage in other social roles that would decrease their developmental burden [11,53].

*Third*, the findings indicate that the adult child's dependence in ADLs only had a weak association with parental perceptions of emotional and financial burden. In the case of financial burden, the association with dependence in ADLs was completely explained by parental health. Other factors, such as the availability of medical insurance and discretionary income to pay for out-of-pocket services, are likely to have a stronger influence on financial burden than the adult child's dependence in ADLs. Similarly, dependence in ADLs contributed only to a small portion of parental emotional burden. Factors beyond the scope of this study, such as depression, quality of life, stress, and anxiety, are likely to have a stronger influence on parental emotional burden. It also is possible for these factors to mediate the association between dependence in ADLs and emotional burden. Characterizing the indirect pathway through these mediators is an interesting line of investigation that should be considered in future research. Despite the wealth of evidence on the impact of behavioral problems of children with ASD on maternal well-being, little is known about the influence of other predictors such as functional independence [68,69]. Therefore, the findings highlight the knowledge gap regarding the influence of dependence in ADLs on caregiver burden experienced by aging parents of adults with ASD.

Although the focus of this study was to illustrate the relationship between dependence in ADLs of adults with ASD and parental perceptions of caregiver burden, it is important to recognize that parenting is joyful and rewarding. A qualitative investigation with a subsample of the study participants revealed that caregiving for adult children with ASD involves both emotional and tangible reciprocity that leads to positive personality changes in both the aging parent and the adult child [26].

## Practice implications

The association of dependence in ADLs of adult children with ASD and parental perceptions of caregiver burden is of practical relevance to family members, practitioners, and policy makers. The findings are relevant to practitioners, such as occupational therapists, social workers, and psychologists, who are committed to improving the quality of life of individuals with ASD and their family members. For example, occupational therapy researchers found that parents often identify independence in ADLs as a top area of intervention for their children with ASD [70]. Gaining mastery in ADLs also is associated with positive quality of life outcomes for adults with ASD, such as post-secondary education, employment, and independent living [15,40,71]. Two practical ways of empowering aging parents of adult children with ASD in dealing with their daily challenges related to ADLs are described below.

First, therapists should consider providing direct services to adolescents and adults with ASD to improve their functional independence. Doing so can decrease the time constraints

faced by parents in helping their adult children and thus free up the time and energy required for engaging with same age peers in leisure and community engagement. It is recommended that older children with ASD be provided with non-traditional direct services to increase affordability and cost-effectiveness. Practitioners should create lifelong, sustainable community-based programs to empower adults with ASD who are at risk for decline in their ADL performance after adolescence [72].

Second, therapists should develop trainings focused on ADLs for families of adults with ASD as has been demonstrated with other caregiver populations [57,73]. For example, aging parents can be trained to help adults with ASD gain and/or maintain mastery in ADL skills into adulthood. Such parental training is important in light of longitudinal research indicating that adults with ASD are at risk to lose mastery over their daily living skills [12,72]. Including siblings of children with ASD in ADL-focused interventions can reduce a culture of dependence in the home environment from childhood and promote independence in ADLs. Further, including siblings in therapy sessions can enhance emotional connections with the child with ASD leading to stronger relationships in adulthood [74,75].

This study also yields important policy implications. In particular, there is a critical need to devise programs focused on addressing the challenges faced by families of adults with ASD who often struggle to find support for their adult children as they transition out of the public-school system [26,41]. Emerging research points to the need for ADL therapies to improve family quality of life of individuals with ASD [66]. The increased incidence of ASD, along with the chasm in service needs for adults with ASD, has created an emerging niche for practitioners such as occupational therapists [55,76]. Therapists should consider community-based models of service delivery because they tend to be family-centered, user-friendly, and cost-effective when compared to the traditional medical-model of service delivery that can lead to long-term dependence on professionals [77,78].

## Limitations

The limitations of this study pertain to the homogeneity of the sample and challenges associated with standardized measurements of ASD traits. The non-probability sampling techniques used to recruit participants resulted in sample homogeneity (*i.e.*, the participants were predominantly well-educated, affluent, married, Caucasian mothers who could access the Internet to participate in this web-based survey). The survey response rate could not be determined, as there was no way of determining how many potential participants heard about the opportunity to participate given the wide spread methods of disseminating the survey link. The overrepresentation of non-Latino white families from a higher socioeconomic status limits the generalizability of the findings. Parents were asked to self-report their adult child's diagnosis of ASD instead of using an autism checklist or a specialized scale to measure the severity of the condition and the level of dependence in ADL. The response categories used for determining the core characteristics of ASD were based on salient criteria available in the literature. The survey design, allowed respondents to choose only one criterion. However, it is possible for individuals to have a range of traits that overlap several responses.

## Future directions

Future research should include a broader array of aging parents of adults with ASD, particularly by targeting the recruitment of racial/ ethnic minorities and those living in low resource, disadvantaged households, and communities. Future efforts should be invested in increasing the accuracy of measurements of the strengths and challenges faced by adults with ASD. This could be done in two ways. First, integrate standardized scales to measure the severity of ASD

including social communication skills and adaptive functioning. Second, researcher-developed survey items included in this study could be refined by asking the participants to rank (on a scale of 0–100) the challenges of their adult child with ASD on each of the response categories within each of the domains (communication, social skills, and behavioral challenges). Doing so, would allow for a more precise characterization and adjustment to these challenges in statistical models. Finally, it is recommended that the CBI and the CRA be validated for aging caregivers of adults with ASD.

## Conclusion

The present study's findings inform gaps in the aging and disability literature on the relationship between the mastery of ADLs by adults with ASD and their aging parents' perceptions of caregiver burden. Promoting functional independence of adult children with ASD has the potential to decrease parental perceptions of burden, specifically that of time dependence and developmental burden that are associated with daily caregiving tasks. In conclusion, findings from this study have the potential to sensitize therapists and policy makers on the impact of daily living skills of adults with ASD on their aging parents' well-being.

## Supporting information

**S1 Fig. Correlations of caregiver burden with independent variables.**
(DOCX)

**S2 Fig. Linear, quadratic, and non-parametric (Lowess) representations of bivariate associations between ADL and caregiver burden.**
(DOCX)

**S3 Fig. Linear associations between ADL and caregiver burden.**
(DOCX)

**S4 Fig. Linear, quadratic, and cubic associations between ADL and caregiver burden.**
(DOCX)

**S5 Fig. Linear associations between ADL dependence and caregiver burden.**
(DOCX)

**S1 Table. Fit Statistics for alternative parametric fit of the association between ADL and caregiver burden outcomes.**
(DOCX)

**S2 Table. Alternative parametric associations, and fit statistics, between ADL and caregiver burden outcomes.**
(DOCX)

**S1 Data.**
(XLS)

**S2 Data. Codebook for analysis.**
(TXT)

## Author Contributions

**Conceptualization:** Preethy Sarah Samuel.

**Data curation:** Christina N. Marsack-Topolewski, Wassim Tarraf.

**Formal analysis:** Preethy Sarah Samuel, Wassim Tarraf.

**Funding acquisition:** Christina N. Marsack-Topolewski, Preethy Sarah Samuel.

**Investigation:** Christina N. Marsack-Topolewski.

**Methodology:** Preethy Sarah Samuel, Wassim Tarraf.

**Project administration:** Christina N. Marsack-Topolewski.

**Resources:** Christina N. Marsack-Topolewski.

**Software:** Wassim Tarraf.

**Supervision:** Preethy Sarah Samuel.

**Validation:** Wassim Tarraf.

**Visualization:** Wassim Tarraf.

**Writing – original draft:** Preethy Sarah Samuel.

**Writing – review & editing:** Christina N. Marsack-Topolewski, Preethy Sarah Samuel, Wassim Tarraf.

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
