## [Decision Letter · Decision Letter 0]

5 Jun 2020

PONE-D-20-02037

Empirical evaluation of the association between daily living skills of adults with autism and parental caregiving burden

PLOS ONE

Dear Dr. Samuel,

Thank you for submitting your manuscript to PLOS ONE. I apologize for the delay. 

After careful consideration, we feel that it has merit but does not fully meet PLOS ONE’s publication criteria as it currently stands. Therefore, we invite you to submit a revised version of the manuscript that addresses the points raised during the review process.

We look forward to receiving your revised manuscript.

Kind regards,

Ece Uzun, PhD

Academic Editor

PLOS ONE

2. Please address the following:

- Please include additional information regarding the survey or questionnaire used in the study and ensure that you have provided sufficient details that others could replicate the analyses. For instance, if you developed a questionnaire as part of this study and it is not under a copyright more restrictive than CC-BY, please include a copy, in both the original language and English, as Supporting Information.

- Please refer to any sample size calculations performed prior to participant recruitment. If these were not performed please justify the reasons. Please refer to our statistical reporting guidelines for assistance (https://journals.plos.org/plosone/s/submission-guidelines.#loc-statistical-reporting).

"This research was funded in part by dissertation research funding from Wayne State University’s Graduate School and the School of Social Work that enabled the data collection."

"No."

Reviewers' comments:

Reviewer's Responses to Questions

**Comments to the Author**

1. Is the manuscript technically sound, and do the data support the conclusions?

Reviewer #1: Yes

Reviewer #2: Yes

2. Has the statistical analysis been performed appropriately and rigorously? 

Reviewer #1: Yes

Reviewer #2: Yes

3. Have the authors made all data underlying the findings in their manuscript fully available?

Reviewer #1: Yes

Reviewer #2: No

4. Is the manuscript presented in an intelligible fashion and written in standard English?

Reviewer #1: Yes

Reviewer #2: Yes

5. Review Comments to the Author

Reviewer #1: Thank you for the opportunity to review the manuscript “Empirical evaluation of the association between daily living skills of adults with autism and parental caregiving burden”. This manuscript addresses an important topic, which is relevant and likely to be of high interest across multiple sectors concerned about the health and well-being of persons with disabilities, including allied health, developmental disabilities, education, welfare, and social services. It is indeed surprising that although the majority of adults with autism live at home with their parent, there is a dearth of on aging parental caregiver burden in this population. One limitation of the study is the extremely modest contribution to the extant literature. The primary finding is that lower adaptive functioning in daily living of individuals with autism was associated with greater perceived caregiver burden. This is finding is well-established across the life course in this population specifically and more broadly with parental caregivers of adults with disabilities.

I appreciated the cogent and conceptually complex literature review in presenting the multifaceted aspects of caregiver burden and its measurement. One limitation of the literature review is its conceptualization of ADL dependence as primarily an intraindividual phenomenon without attention to or coverage of the well-established literature on the crucial role of intervention and supports preceding, during, and following the transition to adulthood for persons with disabilities and specifically with autism. In fact, the lack of such supports is a major contributing factor to the lack of development of independent ADL, which preclude opportunities for independent living.

Measures and analytical approach are appropriate to address the research questions. A limitation of the sampling is that it overrepresents white, higher income families and yet we know that the systemic inequities place individuals who are of non-dominant race and/or who are socioeconomically disadvantaged at higher risk for developmental disabilities.

Suggestions for strengthening this manuscript include the following:

(1) There is a well-established research literature spanning 40 years on Community Living Opportunities for persons with disabilities. It would be helpful to incorporate at least a brief review of this literature into the introduction.

(2) After incorporating the above foundation, it would be helpful to consider and state the relevance of the findings relative to the literature above.

(3) The sampling limitations noted above should be noted as limitations in the Discussion section.

Reviewer #2: Thank you for the opportunity to review this manuscript describing a survey study looking at the relationship between the daily living skills of adults on the autism spectrum and caregiver burden. This study had a reasonably large sample size and presents results that can be informative for the literature. However, I noted several areas in need of improvement.

First, I found that there were weak links in the justification for the study. This was notable in the Abstract and also in the beginning of the introduction. By the end of the introduction, these links were made, but I believe edits to the abstract and beginning of the introduction could greatly strengthen the manuscript. I've further described these in two bullets below.

-In the abstract, the second sentence did not adequately link from the first sentence. It does not read as a clear "Therefore" between these two ideas. It felt like a leap when reading it on its own.

-In the introductory section, I found similar issues with leaps in reasoning that don't quite link. The second paragraph on page 3 starts with a sentence that begins stating that there is a benefit, but then finishes stating that there are detrimental impacts on health. Those detrimental impacts are not described, and instead the next sentence begins to talk about more benefits. The reasoning here was hard to follow. Next, there is a sentence that again makes a big leap "parenting can be bittersweet" and then specifically states the study purpose. I believe there is a need for more clearer and stronger writing in this paragraph to more clearly draw these connections and build the rationale for this project. Of course you provide more information alter in the introduction (which was done effectively), but this paragraph as it is currently written does not work.

Second, I suggest that the authors use the acronym "ADLs" rather than "ADL" for activities of daily living. This will make its representation of a plural much more clear, and is consistent with how AOTA uses this term (see the OTPF). When making this change, the authors should review every sentence in which it is used to make sure there is subject-verb agreement, as this was somewhat inconsistent throughout the paper (currently ADL is used as both a singular and plural noun at different points). For example, the phrase "ADL status" would sound strange as "ADLs status" so those instances should be edited throughout ("independence in ADLs" may be a reasonable edit).

Next, in analyses/results, there is some lack of clarity. How were the correlations run (Pearson? Spearman? some variables were linear and others were considered nominal?)? More importantly, I don't actually view the correlations as useful to include. Many of the magnitudes are incredibly small (many 0.1 and 0.2), yet interpreted as "significant". (Magnitudes are stated to be "small" for emotional and financial burden, but there are many others, for example all were small for compound caregiving). They are also not considered further at all in the paper. I believe it would be sufficient to just state that you examined correlations prior to running the regression models. I didn't find that they added anything to the paper. The figure for them was also a bit confusing with the use of the red square for non-significant correlations (especially when alongside dark orange representing strong positive correlations) and that the x and y axes were not in a consistent order. For the regression results, "beta" is reported but it is not clear whether they are standardized or unstandardized coefficients. I also did not find the regression figures to be critical for the paper. Currently, they seem to be duplicated in the supplemental files (in main paper and supplemental). I feel they could work fine as just supplemental and not in the main paper.

Finally, in the discussion section, there are some points that could be made more effectively. Currently, there is an implication that interventions should be provided to adolescents and adults on the spectrum ONLY for the purpose of reducing caregiver burden. I recognize that this ties best with your results, but it comes across strangely and I believe it would be much more appropriate to also discuss the potential benefits of this for the individuals themselves (there is available literature on this). In the limitations section, the authors state that their lack of diversity is acceptable because they were able to control for participant characteristics. However, this is misleading, since one cannot truly control for things when there is high heterogeneity in the sample. This should be re-stated.

Small points:

-On page 5, the term "independent living" is used, where I believe what is meant is "independence in ADLs" (these are not interchangeable terms)

-On page 7, when describing participant ages, 60 is included in both categories. This should be edited to indicate either 50-59 or 61+, depending on which category 60 year olds were considered in.

-Page 7, typo: "bring" should be "being"

-Page 12, where it says "group (3)" I believe it would be more clear to say "a score of 3"

-sentence spanning pages 13-14 was long and hard to read-suggest editing

-In table 2, I believe you should remove the decimal points in the row reporting the sample size

-page 21, "emerging growing" --only one of these words is needed

-page 24, "minorities" may be better stated as "racial/ethnic minorities"

6. PLOS authors have the option to publish the peer review history of their article (what does this mean?). If published, this will include your full peer review and any attached files.

Reviewer #1: No

Reviewer #2: No

---

## [Author Response · Author response to Decision Letter 0]

23 Jul 2020

Thank you for the chance to resubmit. We have undertaken a major revision and uploaded the marked up manuscript and the response to reviewers. Looking forward to a favorable response.

---

## [Decision Letter · Decision Letter 1]

26 Oct 2020

PONE-D-20-02037R1

Empirical evaluation of the association between daily living skills of adults with autism and parental caregiving burden

PLOS ONE

Dear Dr. Samuel,

Thank you for submitting your manuscript to PLOS ONE. After careful consideration, we feel that it has merit but does not fully meet PLOS ONE’s publication criteria as it currently stands. Therefore, we invite you to submit a revised version of the manuscript that addresses the points raised during the review process.

Both Reviewers found the updated version of the manuscript much improved. However, they still have some valuable recommendations. I am asking you to read the comments carefully and try to address the issues raised. 

We look forward to receiving your revised manuscript.

Kind regards,

Ewa Pisula

Academic Editor

PLOS ONE

Reviewers' comments:

Reviewer's Responses to Questions

**Comments to the Author**

1. If the authors have adequately addressed your comments raised in a previous round of review and you feel that this manuscript is now acceptable for publication, you may indicate that here to bypass the “Comments to the Author” section, enter your conflict of interest statement in the “Confidential to Editor” section, and submit your "Accept" recommendation.

Reviewer #2: (No Response)

Reviewer #3: All comments have been addressed

2. Is the manuscript technically sound, and do the data support the conclusions?

Reviewer #2: Yes

Reviewer #3: Yes

3. Has the statistical analysis been performed appropriately and rigorously? 

Reviewer #2: Yes

Reviewer #3: Yes

4. Have the authors made all data underlying the findings in their manuscript fully available?

Reviewer #2: Yes

Reviewer #3: Yes

5. Is the manuscript presented in an intelligible fashion and written in standard English?

Reviewer #2: Yes

Reviewer #3: (No Response)

6. Review Comments to the Author

Reviewer #2: I found the updated version of this manuscript much improved. The writing was very clear and flowed well. The updates throughout made sense and helped increase clarity. I have a few additional points that I think should be considered prior to publication:

-Please clarify why 50 years of age was selected as the minimum age for participants.

-There are still some issues with demographic response categories. See Table 1. ">60" years should be replaced with the 'greater than or equal to' sign. Similarly, Income levels I believe should be 'less than or equal to $40,000" (or use the symbol) since the next category starts at $40,001.

-I find issues with the questions/wording used for the ASD characteristics. Most notably, the response items are not necessarily mutually exclusive. I understand that this was the tool used in the study, so it cannot be changed at this point, but problems with that measurement should be included as a limitation. It should also be noted in the measurement section that these items were developed by your research team (I assume?).

-I appreciate the clarity on the use of the communication scale as nominal and the use of "communicates needs and wants in a meaningful way" as the reference category. However, please edit so that both mentions of that response option are accurate. In the sentence describing the scale, the word "some" is included and then it is not included when it is indicated as the reference category. Use the exact language that was presented to participants.

-the correlational analyses are not described in the data analysis section (but should be if they are included in the results section)

-on page 21, the word "of" seemed missing in the phrase "availability medical insurance"

Reviewer #3: Overall comments

I appreciate that the authors commented on positives as well as negatives of caring for children on the autism spectrum. I am also pleased to see the recognition of ongoing parental support because this is often neglected.

The adults that I work with prefer to use identity-first language rather than person-first. I therefore refer to my participants as autistic adults, please review the article by Lorcan Kenny (https://doi.org/10.1177/1362361315588200).

When writing I prefer not to start sentences with acronyms, on page 5 the authors do this and I would encourage them to reconsider this. I also prefer scientific writing to use the third-person point of view. However, these are stylistic comments and would not preclude publication.

Introduction

The introduction outlines to purpose and logic behind the study.

The sentence "As individuals with ASD reach adulthood, many continue to require some form of reliance on their parents" needs to reviewed. It does not make sense to "require" "reliance" of someone. In the last sentence on page 4 the authors start a sentence with "because" and this is not an appropriate fashion to start sentences in academic writing - please review.

On page 5, the sentence "Aging parents, who are themselves experiencing age-related health declines while supporting their adult children ..." needs reviewing. Please consider a different phrasing for "health declines".

The authors state the purpose of the study twice. Is this necessary?

Methods/materials

Research design - have the authors been provided with an ethics approval code? If so please include this code.

The authors indicated that 353 surveys were received, but how many potential participants were provided with information regarding the survey. This would inform the reader of your response rate.

Materials used - are the CBI and the CRA valid and reliable to use in this population.

The authors have not included a "procedure" section. This information is included in other aspects of the manuscript, but it would be clearer if there was a stand alone procedure section.

Discussion

In the first paragraph of the discussion the authors indicated that future qualitative research will be completed. I feel this would be more appropriate to include in the 'future research' section of the discussion (just before the conclusion).

On page 21 on line 7, the authors switch to present tense when describing their results. Please change this to past tense.

The first section of the discussion (before practice implications) initially describes the results and then link this to other results. I feel that this could be approached from a more integrated fashion, but I feel this is stylistic.

Conclusion

this is nice and clear

7. PLOS authors have the option to publish the peer review history of their article (what does this mean?). If published, this will include your full peer review and any attached files.

Reviewer #2: No

Reviewer #3: No

---

## [Author Response · Author response to Decision Letter 1]

19 Nov 2020

The MS Word file has bene uploaded. Data file has been uploaded as an MS Excel file and the codebook is a text file.

---

## [Decision Letter · Decision Letter 2]

18 Dec 2020

Empirical evaluation of the association between daily living skills of adults with autism and parental caregiver burden

PONE-D-20-02037R2

Dear Dr. Samuel,

We’re pleased to inform you that your manuscript has been judged scientifically suitable for publication and will be formally accepted for publication once it meets all outstanding technical requirements.

Preparing the final version of the manuscript, please take into account two suggestions of Reviewer #3: 

"Please note that in your abstract you use the abbreviation ADL before writing it in full and this should be corrected

On page 21 in the discussion there is a sentence "Similarly, dependence in ADLs contributed only to a small portion

of parental emotional burden". I believe the word "to" should be removed."

Kind regards,

Ewa Pisula

Academic Editor

PLOS ONE

Additional Editor Comments (optional):

Reviewers' comments:

Reviewer's Responses to Questions

**Comments to the Author**

1. If the authors have adequately addressed your comments raised in a previous round of review and you feel that this manuscript is now acceptable for publication, you may indicate that here to bypass the “Comments to the Author” section, enter your conflict of interest statement in the “Confidential to Editor” section, and submit your "Accept" recommendation.

Reviewer #2: (No Response)

Reviewer #3: All comments have been addressed

2. Is the manuscript technically sound, and do the data support the conclusions?

Reviewer #2: Yes

Reviewer #3: Yes

3. Has the statistical analysis been performed appropriately and rigorously? 

Reviewer #2: Yes

Reviewer #3: Yes

4. Have the authors made all data underlying the findings in their manuscript fully available?

Reviewer #2: Yes

Reviewer #3: Yes

5. Is the manuscript presented in an intelligible fashion and written in standard English?

Reviewer #2: Yes

Reviewer #3: Yes

6. Review Comments to the Author

Reviewer #2: The authors have addressed my prior concerns. The only thing that seems unaddressed is that there is still a discrepancy between levels in Table 1 for parental "Educational Attainment"--"less than high school' and "some college" are presented as the first two categories. I assume that the first category should be "high school or less" or something to that effect. Since there are likely parents who completed high school, but did not go to college. As currently written, there is no category for those folks. I assume "some college" would include Associate's Degrees. I recommend in future studies to make these options more clear for respondents.

Reviewer #3: Thank you for addressing my concerns with this paper.

Please note that in your abstract you use the abbreviation ADL before writing it in full and this should be corrected

On page 21 in the discussion there is a sentence "Similarly, dependence in ADLs contributed only to a small portion

of parental emotional burden". I believe the word "to" should be removed.

The idea of ongoing parental support for autistic adults is important to address. I also strongly support your suggested clinical implications.

7. PLOS authors have the option to publish the peer review history of their article (what does this mean?). If published, this will include your full peer review and any attached files.

Reviewer #2: No

Reviewer #3: No

---

## [Editor Report · Acceptance letter]

26 Dec 2020

PONE-D-20-02037R2 

Empirical evaluation of the association between daily living skills of adults with autism and parental caregiver burden 

Dear Dr. Samuel:

I'm pleased to inform you that your manuscript has been deemed suitable for publication in PLOS ONE. Congratulations! Your manuscript is now with our production department. 

Kind regards, 

on behalf of

Dr. Ewa Pisula 

Academic Editor

PLOS ONE